# Immune-Related Endocrine Dysfunctions in Combined Modalities of Treatment: Real-World Data

**DOI:** 10.3390/cancers13153797

**Published:** 2021-07-28

**Authors:** Wing-Lok Chan, Horace Cheuk-Wai Choi, Patty Pui-Ying Ho, Johnny Kin-San Lau, Rosa Pui-Ying Tse, Joyce Au, Vivian Lam, Ronald Liu, Isaac Ho, Charlotte Wong, Ben Cheung, Eric Lam, Daryn Chow, Ka-On Lam, Kwok-Keung Yuen, Dora Lai-Wan Kwong

**Affiliations:** 1Department of Clinical Oncology, The University of Hong Kong, Hong Kong, China; hcchoi@hku.hk (H.C.-W.C.); lamkaon@hku.hk (K.-O.L.); dlwkwong@hku.hk (D.L.-W.K.); 2Department of Clinical Oncology, Queen Mary Hospital, Hong Kong, China; hpy408@ha.org.hk (P.P.-Y.H.); lks417@ha.org.hk (J.K.-S.L.); tpy804@ha.org.hk (R.P.-Y.T.); als714@ha.org.hk (J.A.); lsm738@ha.org.hk (V.L.); lr638@ha.org.hk (R.L.); hi166@ha.org.hk (I.H.); wcc742@ha.org.hk (C.W.); cmf078@ha.org.hk (B.C.); lkm496@ha.org.hk (E.L.); cdy455@ha.org.hk (D.C.); yuenkk1@ha.org.hk (K.-K.Y.)

**Keywords:** immune checkpoint inhibitors, immune-related endocrine dysfunction, hypothyroidism, targeted therapy, malignancy

## Abstract

**Simple Summary:**

Immune-checkpoint inhibitors (ICI) have been increasingly used in the management of various types of cancers. More studies and guidelines also recommended the combination of ICI with other anti-cancer agents to improve the efficacy and treatment outcome. This retrospective study showed that the combination of ICI and targeted agents increased the risk of immune-related endocrine dysfunction and hypothyroidism. Moreover, older patients on ICI had a higher risk of immune-related diabetes mellitus. ICI is safe and well-tolerated regardless of age, but close monitoring of fasting glucose is essential in older populations.

**Abstract:**

The number of immune-related endocrine dysfunctions (irEDs) has concurrently increased with the widespread use of immunotherapy in clinical practice and further expansion of the approved indications for immune checkpoint inhibitor (ICI) in cancer management. A retrospective analysis was conducted on consecutive patients ≥18 years of age with advanced solid malignancies who had received at least one dose of anti-programmed cell death protein 1 (anti-PD-1) and/or anti-CTLA4 antibodies between January 2014 and December 2019 at a university hospital in Hong Kong. Patients were reviewed up to two months after the last administration of an ICI. The types, onset times and grades of irEDs, including hypothyroidism, hyperthyroidism, adrenal insufficiency and immune-related diabetes mellitus, were recorded. Factors associated with irEDs were identified using multivariate analysis. A total of 953 patients (male: 603, 64.0%; median age: 62.0 years) were included. Of these, 580 patients (60.9%) used ICI-alone, 132 (13.9%) used dual-ICI, 187 (19.6%) used an ICI combined with chemotherapy (chemo + ICI), and 54 (5.70%) used immunotherapy with a targeted agent (targeted + ICI). A significantly higher proportion of patients using targeted + ICI had irEDs and hypothyroidism; in contrast, a higher proportion of patients using dual-ICI had adrenal insufficiency. There was no significant difference in the incidence of irED between the younger (<65 years) and older (≥65 years) patients. Using logistic regression, only treatment type was significantly associated with irEDs. Notably, older patients had a higher risk of having immune-related diabetes mellitus. This large, real-world cohort demonstrates that targeted + ICI has a higher risk of overall irED and hypothyroidism. Immunotherapy is safe and well-tolerated regardless of age, but close monitoring of fasting glucose is essential in older populations.

## 1. Introduction

Immune checkpoint inhibitors (ICIs) have become a powerful tool in the management of cancer in recent years. These monoclonal antibodies (mAb) block immune checkpoints, unleashing T-cells to fight cancer [1]. The ICIs approved for the treatment of different cancers include agents that target the programmed cell death protein 1 receptor (PD-1: nivolumab, pembrolizumab, cemiplimab), programmed death-ligand 1 (PD-L1: atezolizumab, avelumab, durvalumab), and cytotoxic T-lymphocyte-associated protein 4 (CTLA-4: ipilimumab, tremelimumab) [2,3].

Anti-PD-1 or anti-PD-L1 monotherapy has been approved for the treatment of over ten types of cancer and has good safety profile [4]. However, the objective response rate was only around 15–20%. Because of its broad-spectrum anti-tumor activity and good tolerability, a growing number of clinical trials have been investigating the combination of ICIs with other immunomodulatory agents or conventional systemic anti-cancer therapy, including cytotoxic chemotherapy or targeted molecular therapy in order to improve the treatment outcome. Some of these combinations have already demonstrated significant improved clinical outcomes and have been approved by the U. S. Food and Drug Administration (FDA). For example, the combination of pembrolizumab with pemetrexed and carboplatin significantly improved the overall survival (OS) and progression-free survival (PFS) in metastatic non-small cell lung carcinoma [5]; axitinib with pembrolizumab improved the OS and PFS in advanced renal cell carcinoma [6]; nab-paclitaxel and atezolizumab improved in OS and PFS in PD-L1 positive advanced triple-negative breast cancer [7]; bevacizumab and atezolizumab hepatocellular carcinoma resulted in better OS and PFS compared with sorafenib [8], etc.

Immune checkpoints are important in maintaining immunological self-tolerance and preventing autoimmune disorders. ICIs remove self-tolerance, triggering autoimmune adverse events leading to toxicities, termed immune-related adverse events (irAEs) [9]. These irAEs can occur in any organ in the body, causing colitis/diarrhea, dermatitis, hepatitis, renal impairment, endocrinopathies and, less commonly, neuropathy, myocarditis and ocular involvement.

Endocrinopathy is the most common irAE associated with ICI therapy. Immune-related endocrine dysfunctions (irEDs) include hypo-physitis, thyroid dysfunction, insulin-deficient diabetes mellitus and adrenal insufficiency. These irEDs are frequently reported in clinical trials and manageable if detected early [10,11,12]. Yet these can be life-threatening if not recognized or treated appropriately; deaths have been reported.

With the expansion of the approved indications of ICIs and increasing use in clinical practice, the incidence of irED is also growing. It is important to understand irED patterns in order to achieve earlier detection and better monitoring.

Previous meta-analyses revealed that the combination of anti-PD-1 plus anti-CTLA4 had a higher risk of irEDs. However, data from clinical trials often involve patients who are younger and fitter than in the real-world situation. Moreover, the pattern of irED is unclear when ICIs are combined with other treatment modalities.

Our study aims to evaluate the incidence and patterns of different irEDs and compares them across different combination treatment modalities. Secondly, we aim to compare the irED profiles in older patients with those in younger patients as older populations are often under-represented in clinical trials and the safety of ICIs in this population has not been adequately assessed.

## 2. Methods

### 2.1. Study Design, Setting, Samples

We retrospectively analyzed data on consecutive patients aged ≥18 years with advanced solid malignancy who had received at least one dose of anti-PD-1 and/or anti-CTLA4 antibodies with or without combined chemotherapy or targeted agent between January 2014 and December 2019 at Queen Mary Hospital, a university hospital and tertiary oncology center in Hong Kong. Patients diagnosed with hematologic malignancies were excluded. Inclusion criteria were: (1) a histologically confirmed diagnosis of solid malignancy; (2) locally advanced or metastatic disease; (3) age ≥18 years at ICI initiation; (4) receipt of at least one cycle of ICI. ICI studied included antibodies targeting PD-1 (nivolumab and pembrolizumab), PD-L1 (atezolizmab, durvalumab) and CTLA-4 (ipilimumab).

### 2.2. Data Collection

All patient data were extracted from the Clinical Management System (CMS) under the Hospital Authority. The following clinical, biological and laboratory data were captured at baseline: (a) age, gender, Eastern Cooperative Oncology Group-Performance Status (ECOG-PS); (b) primary tumor site, site of metastasis, cancer type and histological subtype, site of metastasis; (c) immunotherapy used: type of ICI, treatment start date, concomitant anti-cancer treatment, previous use of ICI; (d) baseline serum endocrine blood tests.

### 2.3. Endpoint Definition

The primary endpoint was immune-related endocrine dysfunction (irED). irEDs were recorded and reviewed by the principal investigator up to two months after the last administration. The types of irED captured included hypothyroidism, hyperthyroidism, adrenal insufficiency and immune-related diabetes mellitus. The date of onset, duration after starting ICIs and grade were collected. The data cut-off was 31 July 2020, and data were censored at patients’ last documented clinic visit.

### 2.4. Statistical Analysis

Patients were categorized into four groups: use of immunotherapy alone (ICI-alone), dual immune-checkpoint inhibitors (dual-ICI), ICI with chemotherapy (chemo + ICI) and ICI with targeted therapy (targeted + ICI). The population was also categorized into younger and older age groups according to age at treatment initiation: <65 years and ≥65 years old.

Descriptive analyses were used to summarize study sample characteristics and endocrinopathy data. The proportion of endocrinopathy was compared across age categories using the Pearson X2, or Fisher-exact test. Univariable analyses were performed to examine the association between age group, site of metastasis, concomitant use of other systemic anti-cancer treatment and irED.

Multivariable analyses were performed to determine those variables significantly contributing to the irED. Age group, site of metastasis and concomitant use of other systemic anti-cancer therapy were used as independent variables. The odds ratios for the independent variables were generated by logistic regression. All reported *p*-values were two-sided, and the significance threshold was set at at <0.05. All statistical analyses were performed using SPSS, version 26 (IBM).

## 3. Results

Baseline clinical characteristics of the patients are shown in Table 1.

A total of 953 eligible patients were identified. 610 patients (64.0%) were male. The median age of the patients was 62.0 years, and 218 patients were in the older age group. The median age of patients in the older age group was 71.9 years (range: 65–103), whereas that in the younger age group was 54.7 years (range: 20–65). The percentage use of ICI-alone, dual-ICI, chemo + ICI and targeted + ICI was 60.9%, 13.9%, 19.6% and 5.70%, respectively. All patients in the dual-ICIs group received combination of nivolumab and ipilimumab. The types of immunotherapy used are shown in Table 2.

### 3.1. Overall Immune-Related Endocrine Dysfunction

A total of 279 patients (29.3%) experienced any type of irED. None of the patients had life-threatening conditions or death due to irED. A significantly higher proportion of patients in the targeted + ICI group had irED (*n* = 25, 46.3%, *p* = 0.002) compared with ICI-alone, dual-ICIs and chemo + ICI groups. In both older and younger age groups, patients who received targeted + ICI had a significantly higher percentage of irED compared with ICI alone. Details on the irED in each subgroup are listed in Table 3.

### 3.2. Thyroid Dysfunction:

Hypothyroidism (*n* = 171, 17.9%) is the most common irED. The incidence of hypothyroidism was significantly higher in the targeted + ICI group (35.2%), compared with that in other groups (ICI-alone: 16.9%; dual-ICIs: 22.0%; chemo + ICI: 13.4%). In both the younger and older age groups, the percentage of hypothyroidism was also significant in the targeted + ICI group (older: 45.5%; younger: 32.6%). The overall median time of onset of hypothyroidism was 12.55 weeks. The onset time was earlier in the targeted + ICI group (median onset: 9.75 weeks) compared with other groups.

Two patients had grade 3 hypothyroidism which was complicated with hyponatremia. Both patients received targeted + ICI and were in the younger age group. They needed temporary treatment discontinuation and hospitalization; subsequently, their condition improved, and ICI treatment was resumed.

Fifty-nine patients (6.19%) had hyperthyroidism and needed antithyroid medications. None of them had grade 3 or above hyperthyroidism. There was no significant difference in percentage with or without concomitant anti-cancer agents. The incidence of hyperthyroidism in both age groups was comparable. The median time of onset of hyperthyroidism was 11.4 weeks, with the earliest onset seen in the dual-ICI group (4.59 weeks).

### 3.3. Adrenal Insufficiency

Adrenal insufficiency was the second most common irED (*n* = 71, 7.45%). There was a significantly higher rate of adrenal insufficiency in the dual-ICI group (14.4%) compared with the ICI-alone group. The median time of onset of adrenal insufficiency was 18.7 weeks. The incidence of adrenal insufficiency was higher in the younger age group than the older age group (8.57% vs 3.67%, *p* = 0.023). In the younger age group, significantly higher incidence of adrenal insufficiency was seen in the dual-ICI group (16.5%) compared with other groups. Three patients had grade 3–4 adrenal insufficiency. Two patients (aged 50 and aged 64) receiving chemo + ICI had sepsis and found adrenal insufficiency. One patient (aged 70) on dual-ICI experienced severe malaise, and blood tests reported adrenal insufficiency.

### 3.4. Immune-Related Diabetes Mellitus

Twenty-eight patients (2.90%) had hyperglycemia while on ICI. There was no significant difference in incidence between the four treatment groups. The median onset of hyperglycemia was 19.6 weeks. The incidence was significantly higher in the older age group (*n* = 14, 6.40%) than the younger age group (*n* = 14, 1.90%). None of the patients had diabetes ketoacidosis or needed hospitalization due to hyperglycemia alone.

### 3.5. Factors Associated with irED

Logistic regression was performed to assess the relative influence of age category, treatment type and sites of metastasis on the risk of irED (Table 4). With both univariable and multivariable analysis, only treatment modality was significantly associated with overall irEDs. Patients receiving targeted + ICI had a higher risk of irED toxicities (OR: 2.34; 95% CI: 1.32–4.41 for targeted + ICI relative to ICI alone).

Further analyses were performed on each type of irED. Patients who received targeted + ICI had a higher risk of hypothyroidism compared with those who received ICI alone (OR: 2.54; 95% CI: 1.36–4.61). Patients on dual-ICI and those in the younger age group had a higher risk of adrenal insufficiency compared with those with ICI-alone (OR: 3.25; 95% CI: 1.70–5.99) and those in the older age group (OR: 0.39, 95% CI: 0.17–0.80), respectively. On the other hand, the older age group had a higher risk of immune-related diabetes compared with the younger age group (OR: 3.78; 95% CI: 1.73–8.32). Sex and sites of metastasis were not related to the risk of irED.

Concerning the type of targeted agent associated with higher risk of irED, none of them showed significant differences. Table 5 showed the number and percentage of irED and hypothyroidism of different types of targeted agents. The targeted agents included in this study is shown in Appendix A.

## 4. Discussion

Immune checkpoint inhibitors are undoubtedly a breakthrough in cancer therapy in recent years and have been widely used in the management of various cancers. To further improve the treatment outcomes, combinations of immunotherapy with other systemic anti-cancer treatments have been investigated in various cancer types and have been increasingly and extensively used. While the main goal is to improve overall survival and maintain quality of life, the implicated toxicity of these combined treatment modalities should also be carefully monitored.

Previous studies focused mainly on irAEs after combination immunotherapies or combination of immunotherapy with chemotherapy. Zhang’s meta-analysis, which involved 11 randomized controlled trials (RCTs) and 5207 patients, demonstrated that combination immunotherapies had a higher risk of irAEs, with ratios of all-grade diarrhea of 1.95 (95% CI: 1.54–2.46; *p* < 0.00001) and all-grade colitis of 4.45 (95% CI: 3.04–6.51; *p* < 0.00001) [13]. Sousa’s study, which involved 38 RCTs comprising 7551 patients, found that patients on combination immunotherapies were significantly more likely to experience hypothyroidism and hyperthyroidism than those on mono-immunotherapy [14]. Another study by Carretero-González et al., which involved 10 RCTs and 4379 patients, revealed that patients receiving combination of immunotherapy with chemotherapy had more grade 3/4 adverse events (RR: 1.32; 95% CI: 1.12–1.55) and discontinuations (RR: 2.31; 95% CI: 1.28–4.16) [15].

Our study observed significantly higher incidences of any irEDs and hypothyroidism in patients treated with targeted + ICI compared with ICI-alone. Additionally, the onset of hypothyroidism was significantly earlier in patients who received targeted + ICI. These findings are important, as the combination of immunotherapy with targeted therapy has been increasingly used, and many phase II and III studies are now investigating the efficacy of combining the two anti-cancer modalities in both solid cancers and hematologic malignancies. For example, from clinictrials.gov, currently there are already over 50 phase II/III studies investigating the use of ICIs with either chemotherapy or targeted agents.

Some of the targeted agents, particularly tyrosine kinase inhibitors targeting VEGFR1–3 (Vascular Endothelial Growth Factor Receptor) or PDGFR (Platelet Derived Growth Factor Receptor), are well known to cause thyroid dysfunction. For example, the incidence of hypothyroidism of sunitinib ranges from 50% to 80% while that of sorafenib ranges from 20% to 35% [16]. The probability of developing TKI-induced thyroid dysfunction depends on patient’s background (higher risk in female and older patients), presence of associated thyroid disorder, the duration of the use of targeted agents, and the molecule.

Suggested mechanisms for targeted-agent-induced hypothyroidism include direct toxic effects on thyrocytes, reduced TPO activity, impaired iodine uptake, attenuation of thyroid blood flow due to vascular epithelial growth factor receptor inhibition, and activation of cytotoxic T cells in combination with pre-existing intrathyroidal lymphocytes causing damage to the thyroid cells [17].

Iatrogenic thyroid disorder related to ICI is also well documented in previous studies, with prevalence of 3.1%–25% in patients treated with ICI as monotherapy. The pathophysiological conception of thyroid dysfunction linked to ICI is still uncertain. Different hypothesis proposed include the overexpression of HLA-DR causing monocytes activation and infiltration into the thyroid tissues, cytotoxic T cells causing destructive thyroiditis and weakened immune tolerance due to overexpression of PD-L1/PD-L2 [18].

The two different pathways from targeted agents and ICI probably exacerbate the risk of thyroid dysfunction. However, due to the diverse types of targeted agents involved in this study as well as relatively small number of patients who used targeted + ICI, no particular type of targeted agent when combined with ICI was identified to have higher incidence of irED. Larger scale studies, e.g., meta-analysis with involvement of RCTs, may give a better answer on the type of targeted agents that have a higher risk of irEDs. More in-vitro and in-vivo studies are warranted to investigate the mechanism of endocrine dysfunction with the combination of treatment.

Results from previous studies investigating the association between age and irAEs were conflicting. Betof’s study demonstrated an increased incidence of irEDs and hypothyroidism in older patients with melanoma who underwent immunotherapy [19]. Baldini’s study showed that the incidence of grade 2 or above irAEs was higher in patients over 70 [20]. On the other hand, Sattar’s study demonstrated that patients ≥75 years of age did not experience excess toxicity and concurrently had similar benefits from immunotherapy as younger patients [21]. Samani’s study even showed that there was a lower incidence of endocrine toxicity in the older patients (age ≥ 75) compared with younger patients (age < 65) [22].

Our study showed no significant difference in overall irEDs between the younger and older age groups. We suggest that the endocrine monitoring in younger and older populations who are on immunotherapy should not be different.

There was a higher incidence of immune-related diabetes mellitus in our cohort’s older patient group. Immune-related diabetes mellitus is rare but potentially life-threatening. Patients should be monitored for blood glucose level and any signs of diabetic ketoacidosis, which often presents with nausea, vomiting, abdominal pain, hyperventilation, lethargy, and/or coma. Older patients on ICIs should have regular blood glucose checks (e.g., fasting glucose every 3–4 weeks). If a high glucose level is detected, intensive insulin treatment, anti-hyperglycemic medications and supportive measures, including hydration and correction of electrolytes, should be administered.

Most of the irEDs in this study were at grade 1–2. We have summarized the grade G3-4 irAE other than endocrine dysfunction in Appendix A. However, the types of irAE were very heterogeneous and the number of patients suffered from each type of irAE was small. It is thus difficult to interpret if the type of treatment would have any significant association with these irAEs.

Our study has several limitations. First, this is a retrospective study conducted in a single institution and therefore may not be generalizable. Second, irED is our focus in this study and not all irAEs (grade 1–2) were reported. Third, a variety of different types of targeted agent were used concurrently with ICI in this study and the number of patients who used same type of targeted therapy was small. Therefore, it would be difficult to identify any particular type of targeted agents more prone to endocrinopathies when combined with ICI. Despite these weaknesses, our study has several strengths. We included a large cohort of patients with different tumor types and treatment types. There was a large proportion of patients (40.0%) who used combined modalities of treatment. The combination of ICIs with chemotherapy or targeted agents is now increasingly used to treat different cancers. In addition, more than one-third of our patients were over 65 years old. This allowed us to have a good comparison of irEDs between the older and younger populations. Moreover, data on different irEDs were meticulously collected.

## 5. Conclusions

Our study demonstrated a higher risk of irEDs in patients who received targeted + ICI. The incidence of hypothyroidism was higher and the onset of hypothyroidism was earlier in targeted + ICI treatment patients. Prospective studies are warranted to better capture irEDs in patients using ICIs combined with other treatment modalities. Future prospective studies and clinical trials could lend more solid evidence and suggest mechanisms for the observed higher incidence. Future research on biomarkers may shed light on the mechanisms and predictions on irAEs. Although older patients are usually frailer and have a higher chance of getting treatment-related toxicities in chemotherapy and targeted therapies than younger ones, our study did not show increased endocrine toxicities in the former group. Research should also focus on whether geriatric assessments or geriatric valuables can better predict both outcomes and toxicities.

## Figures and Tables

**Table 1 cancers-13-03797-t001:** Clinical characteristics of the patients.

Clinical Characteristics	Number	Percentage
Total number of patients	953	
Sex Male Female	610 343	64% 36%
Median age	61.49	Range: 20.1–102.4
Types of cancer Lung GI: Colon/stomach/esophagus HBP: Liver/pancreas/biliary tract Uro: Kidney/bladder/renal pelvis Gynae: ovary/cervix/uterine Head and neck Breast Others	310 69 315 61 15 56 52 75	32.5% 7.24% 33.05% 6.40% 1.57% 5.88% 5.56% 7.87%

**Table 2 cancers-13-03797-t002:** Types of immunotherapy used.

Title	Number	Percentage
Pembrolizumab	552	57.90%
Nivolumab	309	32.40%
Atezolizumab	77	8.10%
Durvalumab	15	1.60%
Ipilimumab	132	13.90%

**Table 3 cancers-13-03797-t003:** Immune-related endocrine dysfunction.

Treatment Modality	Overall				Age > 65			Age < 65		
	Total Number of Patients	Total Number of Event	Percentage	Median Time of Onset (Weeks)	Total Number of Event	Percentage	Median Time of Onset (Weeks)	Total Number of Event	Percentage	Median Time of Onset (Weeks)
Any irED										
Total	953	279	29.28%		104	30.32%		175	28.69%	
ICI-alone	580	155	26.72%		66	27.97%		89	25.87%	
Dual-ICI	132	46	34.85%		11	29.73%		35	36.84%	
Chemo + ICI	187	53	28.34%		18	33.96%		35	26.12%	
Targeted + ICI	54	25	46.30%		9	52.94%		16	43.24%	
Hypothyroidism										
Total	953	171	17.94%	12.55	65	18.95%	11.72	106	17.38%	13.22
ICI-alone	580	98	16.90%	12.34	42	17.80%	12.00	56	16.28%	12.64
Dual-ICI	132	29	21.97%	15.12	7	18.92%	17.12	22	23.16%	14.52
Chemo + ICI	187	25	13.37%	12.31	8	15.09%	10.11	17	12.69%	13.35
Targeted + ICI	54	19	35.19%	9.75	8	47.06%	7.13	11	29.73%	13.31
Hyperthyroidism										
Total	953	59	6.19%	11.36	23	6.71%	12.84	36	5.90%	11.75
ICI-alone	580	32	5.52%	13.54	12	5.08%	16.01	20	5.81%	12.15
Dual-ICI	132	8	6.06%	4.59	4	10.81%	4.32	4	4.21%	4.86
Chemo + ICI	187	15	8.02%	9.13	6	11.32%	10.50	9	6.72%	8.22
Targeted + ICI	54	4	7.41%	15.75	1	5.88%	24.71	3	8.11%	12.76
Primary adrenal insufficiency										
Total	953	71	7.45%	26.01	16	4.66%	27.88	55	9.02%	22.96
ICI-alone	580	32	5.52%	28.29	12	5.08%	27.08	20	5.81%	29.02
Dual-ICI	132	19	14.39%	25.08	2	5.41%	31.21	17	17.89%	24.35
Chemo + ICI	187	16	8.56%	26.25	2	3.77%	29.36	14	10.45%	25.81
Targeted + ICI	54	4	7.41%	11.25	0	0.00%	0.00	4	10.81%	11.25
Immune-related hyperglycemia										
Total	953	28	2.94%	19.63	19	5.54%	18.29	9	1.48%	22.48
ICI-alone	580	21	3.62%	21.63	13	5.51%	20.30	8	2.33%	23.80
Dual-ICI	132	0	0.00%	0.00	0	0.00%	0.00	0	0.00%	17.14
Chemo + ICI	187	7	3.74%	13.63	6	11.32%	13.93	1	0.75%	11.86
Targeted + ICI	54	0	0.00%	0.00	0	0.00%	0.00	0	0.00%	0.00

**Table 4 cancers-13-03797-t004:** Univariable and multivariable analysis on factors associated with immune-related endocrine dysfunction.

	Any ir-ED		Hypothyroidism		Primary Adrenal Insufficiency		Immune-Related Hyperglucemia	
	Univariable	Multivariable	Univariable	Multivariable	Univariable	Multivariable	Univariable	Multivariable
Age category								
<65 (ref)	Ref	Ref	Ref	Ref	Ref	Ref	Ref	Ref
≥65	0.92 (0.66, 1.28)	0.94 (0.66, 1.31)	1.04 (0.69, 1.52)	1.07 (0.71, 1.58)	**0.41 (0.18, 0.81)**	**0.39 (0.17, 0.80)**	**3.53 (1.64, 7.60)**	**3.78 (1.73, 8.32)**
Treatment category								
ICI alone (ref)	Ref	Ref	Ref	Ref	Ref	Ref	Ref	Ref
Dual ICI	1.47 (0.98, 2.19)	1.47 (0.97, 2.19)	1.38 (0.86, 2.18)	1.32 (0.81, 2.10)	**2.88 (1.55, 5.22)**	**3.25 (1.72, 5.99)**	NE	NE
Chemo + ICI	1.08 (0.75, 1.56)	1.08 (0.74, 1.55)	0.76 (0.46, 1.20)	0.75 (0.46, 1.20)	1.60 (0.84, 2.95)	1.55 (0.81, 2.87)	1.04 (0.40, 2.36)	1.14 (0.76, 1.78)
Targeted + ICI	**2.36 (1.34, 4.16)**	**2.34 (1.32, 4.14)**	**2.67 (1.44, 4.81)**	**2.54 (1.36, 4.61)**	1.37 (0.40, 3.63)	1.50 (0.43, 4.06)	NE	NE
Site of metastasis								
Liver	1.08 (0.80, 1.44)	1.00 (0.74, 1.35)	1.31 (0.93, 1.84)	1.24 (0.87, 1.76)	0.62 (0.35, 1.06)	**0.51 (0.28, 0.88)**	0.89 (0.38, 1.94)	1.21 (0.51, 2.71)
Lung	0.91 (0.69, 1.20)	0.92 (0.69, 1.22)	0.96 (0.69, 1.33)	0.97 (0.69, 1.36)	0.69 (0.41, 1.12)	0.68 (0.41, 1.13)	1.16 (0.54, 2.47)	1.15 (0.53, 2.49)
Adrenal/ kidney	0.95 (0.48, 1.79)	1.00 (0.50, 1.91)	0.67 (0.25, 1.50)	0.69 (0.26, 1.55)	0.86 (0.20, 2.44)	1.13 (0.27, 3.32)	0.72 (0.04, 3.52)	0.47 (0.03, 2.36)
Gender								
Female (ref)	Ref		Ref		Ref		Ref	
Male	0.81 (0.61, 1.08)	0.85 (0.72, 1.21)	0.90 (0.64, 1.27)	0.93 (0.67, 1.32)	0.55 (0.34, 0.90)	0.55 (0.33, 1.02)	1.42 (0.64, 3.46)	1.44 (0.65, 3.42)

Bold numbers are significant values.

**Table 5 cancers-13-03797-t005:** Number and percentage of irED and hypothyroidism of different types of targeted agents concurrently used with ICI.

	No. of Patients	No irED	Had irED	*p*-Value	No Hypothyroidism	Had Hypothyroidism	*p*-Value
Total	54	29 (53.7%)	25 (46.3%)	0.967	35 (64.8%)	19 (35.2%)	0.256
Tyrosine kinase inhibitor	31	16 (51.6%)	15 (48.4%)		18 (58.1%)	13 (41.9%)	
vascular endothelial growth factor receptor (VEGFR) inhibitor	11	6 (54.5%)	5 (45.5%)		7 (63.6%)	4 (36.4%)	
Poly (ADP-ribose) polymerase (PARP) inhibitors	2	1 (50.0%)	1 (50%)		1 (50%)	1 (50%)	
Others	10	6 (60.0%)	4 (40%)		9 (90%)	1 (10%)	

## Data Availability

The data presented in this study are available on request from the corresponding author. The data are not publicly available due to patient’s privacy.

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
