# Peer review of "Immune-Related Endocrine Dysfunctions in Combined Modalities of Treatment: Real-World Data"

_cancers, 2021, doi:10.3390/cancers13153797_

Round 1

Reviewer 1 Report

In this monocentric retrospective study, the authors investigate the types, onset times and grades of immune-related endocrine dysfunctions irED, including hypo-thyroidism, hyperthyroidism, adrenal insufficiency and immune-related diabetes mellitus, in a series of patients treated with immune checkpoints inhibitor (ICI) alone; dual ICI; ICI plus chemotherapy; ICI plus targeted therapy. In this large, real-world cohort the authors found that combining ICI with targeted therapy has a higher risk of overall irED and hypothyroidism.  Moreover, they found no significant difference in the incidence of irED between the younger (<65 years) and older (>65 years) patients, even if older patients had a higher risk of having immune-related diabetes mellitus. The topic is interesting and relevant in the fields. This article may be interesting to a wide circle of researchers and physicians, since it gives new information about a specific setting of immune-related adverse events, those about endocrine dysfunctions. In my opinion this could be reported as a strength point of the manuscript and not as a limitation. The manuscript is well written and comprehensively described, it gives an overview of the latest findings and the references were used properly.

I have only minor comment to be addressed: the authors should make the text smoother and carefully revise it since there are many typos.

Author Response

Dear Reviewer,

Thank you very much for your time and effort in reviewing our manuscript. 

We have included over 950 patients who used immune-checkpoint inhibitors and a large number of patients were older adults (aged >65 years old). Our study showed the risk of immune-related endocrine dysfunctions was higher in patients who used targeted agents concurrently with immune-checkpoint inhibitors.

We have done English editing by native English speakers and have checked all typos and grammar.

We sincerely wish our article can be accepted and published in this journal.

Thank you very much.

Best regards,

Dr. Wendy Chan Wing Lok

Clinical Assistant Professor

Department of Clinical Oncology

The University of Hong Kong

Reviewer 2 Report

In the manuscript, 'Immune-related Endocrine Dysfunctions in Combined Modalities of Treatment: Real-world data', authors did a great and long term hard work throughout the manuscript. But it lacks many things and among that, main factor is novelty. 

Authors also mentioned the limitations of the work and the study conducted in a single institution that limits the case reports. And mainly authors did not analyze the risk of combination therapy which is a severe drawback of the study.

With these limitations, it is not possible to reflect the strength of the study as it already lacks a novelty. And I feel that, this manuscript is not eligible for the publication in 'Cancers'. 

Author Response

Dear Reviewer,

Thank you very much for your time and effort in reviewing our manuscript. 

Our study focused on the immune-related endocrine dysfunction (irED) caused by immune-checkpoint inhibitors and their combination with other treatment modalities.  Although there are existing publications on irED, the comparison of different modalities of treatment, especially with combination of targeted therapy, is limited.  Our study involved a large number of patients from Asia and data on different irEDs were also meticulously collected. 

One of our limitations in this study was that it is conducted in a single institution. However, our study involves over 900 patients who used immune-checkpoint inhibitors and around 40% of our participants were using combination therapy. This gave a strong evidence to support the significant association of irEDs with the type of treatment modality (i.e. ICI + targeted agent).

Since our study mainly focuses on irED, we did not capture the immune-related adverse events (irAE) which were only grade 1-2. However, we did capture the irAE which were grade 3-4 in our cohort. The result was presented in Supplementary Table 2.

We have used our big effort and have made major editing on the original article. We sincerely wish our article can be accepted and published in this journal.

Thank you very much.

Best regards,

Dr. Wendy Chan Wing Lok

Clinical Assistant Professor

Department of Clinical Oncology

The University of Hong Kong

Reviewer 3 Report

The manuscript by Chan et al. describes a single-institution retrospective study analyzing the appearance of immune-related endocrine dysfunctions (irEDs) in solid cancer patients treated by the immune checkpoint inhibitors (ICI), either alone or in combination with other agents. While the number of the analyzed cases is noteworthy for a single-center study, the approach on its own it not very novel. There has been a number of reports on this subject.  An interesting information is, however, the considerably higher frequency of irEDs in patients treated with ICI+targeted agents. Therefore, it may be of value to provide more information regarding this group, e.g., what types of targeted agents were applied and whether there is a correlation of irEDs with any particular type of agents, especially as the Authors claim that "some targeted agents may have a higher risk of causing endocrine dysfunction than others." Also, the potential mechanism(s) behind the increase of the irEDs risk in this group should be discussed in more detail.

Author Response

Dear Reviewer,

Thank you very much for your time and effort in reviewing this article.

Our study focused on the immune-related endocrine dysfunction (irED) caused by immune-checkpoint inhibitors and their combination with other treatment modalities.  Although there are existing publications on irED, the comparison of different modalities of treatment, especially with combination of targeted therapy, is limited.  Our study involved a large number of patients from Asia and data on different irEDs were also meticulously collected.  In this revision, we have added the number and percentage of different targeted therapies used (Supplementary Table 1) and analysed the association of different types of targeted agents with each irED (Table 5). However, since the types of targeted agents used were diverse and the number of patients in each group was small, specific types of targeted agent more prone to irED could not be identified in this study. Even with this limitation, this study result still gives an introduction to future studies and guides on further research development on the mechanism of irEDs when targeted agents are used together with immune-checkpoint inhibitors.  We have done literature search on the possible mechanisms for irEDs and hypothyroidism in patients who are on immune-checkpoint inhibitors and targeted agents. This was discussed in the Discussion part.

We understood there are limitations in our study. Nevertheless, this study did show higher incidence of irEDs and hypothyroidism in patients who used targeted therapies concurrent with ICI.  Also the risk of irED was not different between the younger and older cancer patients but older patients had a higher risk of immune-related diabetes mellitus which could be life-threatening if not timely recognized.

We sincerely wish that our article can be published in your journal.

Thank you very much.

Best regards,

Dr. Wendy Chan Wing Lok

Clinical Assistant Professor

Department of Clinical Oncology

The University of Hong Kong

Round 2

Reviewer 2 Report

The authors have thoroughly answered all questions and solved the issues raised. I have no further concerns.